# Alzheimer’s Disease: Treatment Strategies and Their Limitations

**DOI:** 10.3390/ijms232213954

**Published:** 2022-11-12

**Authors:** Elodie Passeri, Kamil Elkhoury, Margaretha Morsink, Kerensa Broersen, Michel Linder, Ali Tamayol, Catherine Malaplate, Frances T. Yen, Elmira Arab-Tehrany

**Affiliations:** 1LIBio Laboratory, University of Lorraine, 54505 Vandoeuvre-lès-Nancy, France; 2UR AFPA Laboratory, Qualivie Team, University of Lorraine, 54505 Vandoeuvre-lès-Nancy, France; 3Department of Applied Stem Cell Technologies, TechMed Centre, University of Twente, 7500AE Enschede, The Netherlands; 4Department of Biomedical Engineering, University of Connecticut Health Center, Farmington, CT 06030, USA

**Keywords:** Alzheimer’s disease, therapeutics strategies, polyunsaturated fatty acids, blood–brain-barrier, liposomes, exosomes, intranasal administration

## Abstract

Alzheimer’s disease (AD) is the most frequent case of neurodegenerative disease and is becoming a major public health problem all over the world. Many therapeutic strategies have been explored for several decades; however, there is still no curative treatment, and the priority remains prevention. In this review, we present an update on the clinical and physiological phase of the AD spectrum, modifiable and non-modifiable risk factors for AD treatment with a focus on prevention strategies, then research models used in AD, followed by a discussion of treatment limitations. The prevention methods can significantly slow AD evolution and are currently the best strategy possible before the advanced stages of the disease. Indeed, current drug treatments have only symptomatic effects, and disease-modifying treatments are not yet available. Drug delivery to the central nervous system remains a complex process and represents a challenge for developing therapeutic and preventive strategies. Studies are underway to test new techniques to facilitate the bioavailability of molecules to the brain. After a deep study of the literature, we find the use of soft nanoparticles, in particular nanoliposomes and exosomes, as an innovative approach for preventive and therapeutic strategies in reducing the risk of AD and solving problems of brain bioavailability. Studies show the promising role of nanoliposomes and exosomes as smart drug delivery systems able to penetrate the blood–brain barrier and target brain tissues. Finally, the different drug administration techniques for neurological disorders are discussed. One of the promising therapeutic methods is the intranasal administration strategy which should be used for preclinical and clinical studies of neurodegenerative diseases.

## 1. Introduction

Neurodegenerative diseases (ND) represent a major current health challenge due to the aging and the lifestyle of the population, the number of people affected, the impact of these diseases on the life of the patients and their caretakers, and the financial impact that these entails [1,2]. Worldwide, more than 50 million people are affected by neurodegenerative diseases, and this number will almost triple to 152 million in 2050 if no effective preventive or therapeutic solutions are found. Alzheimer’s disease (AD) is considered the most frequent type of neurodegenerative disease, occurring in 60% to 80% of all cases [3].

AD, discovered in 1907, has multiple etiologies, but the exact causes of the disease have not yet been clearly established. In addition, no curative treatment has been developed, even more than a century later. There are two forms of AD: (1) the genetic form or the autosomal-dominant AD (ADAD), which occurs before the age of 65, representing less than 1% of cases; (2) the sporadic form. Sporadic Alzheimer’s Disease (SAD) usually occurs after age 65, and the risk of the disease’s occurrence doubles every five years [4]. Here, our review focuses on sporadic AD.

AD pathology results from both structural and functional damage in the central nervous system (CNS), including abnormal aggregation of proteins in the nervous system and neurodegenerative processes. Indeed, two types of lesions have been identified in AD, including amyloid plaques composed of beta-amyloid peptides (Aβ), which accumulate abnormally outside the nerve cells [5], and neurofibrillary tangles (NFT) due to the hyper-phosphorylated tau protein, which accumulates in neurons [6]. AD can be considered a progressive process of biochemical, neurophysiological, neuroanatomical, and cognitive disorders [7]. The initial oligomerization of soluble Aβ in the brain causes localized dysfunctions of dendrites, axonal processes, and synapses.

During the past decade, research efforts have focused on soluble Aβ oligomers (AβO), which appear to be a more toxic and disease-relevant form of Aβ [8,9]. AβO are considered to be pathological agents that appear before the first neuropathological signs of AD [10,11]. Then, brain lesions gradually form, associated with neuronal loss in certain regions of the brain, but without clinical expression [12,13,14,15]. With time, AD is manifested by loss of memory and cognitive abilities [16].

Many therapeutic strategies have been explored for several decades in clinical trials, but the treatments currently available are primarily treatments of symptoms [17,18,19] rather than actual curative therapies. Because of this, attention has turned towards prevention or reducing AD risk. Research has shown that in the world, more than 30% of AD cases could be due to modifiable risk factors, which could provide interesting and promising targets for prevention strategies to reduce the risk of AD-related cognitive decline and perhaps ND in general [3,20,21]. Current challenges focus on improving the early detection of the disease at the preclinical stage [22].

Various studies have shown the critical role of lipid nutrients in the brain and cognitive functions. The brain has significant levels of omega-3 polyunsaturated fatty acids (n-3 PUFA), incorporated into phospholipids, amphiphilic molecules that are components of neuronal membranes [23,24]. Several studies have demonstrated their potential neuroprotective role in neuron function and synapse plasticity [25]. Since diet is an important source of these PUFA, nutritionally-based prevention strategies can be used towards optimizing CNS lipid status [26].

Another challenge in AD treatments is the efficient delivery and targeting of molecules of interest to the brain due to the protective barriers of the central nervous system [27,28]. With the brain as the target tissue, one obstacle that needs to be overcome is the blood–brain barrier (BBB). BBB protects the CNS from potential neurotoxins, toxic organisms, and chemical substances in the blood, but that also limits its accessibility for many therapeutic drug molecules [27].

The use of nanoparticles (NP), with sizes between 10 and 1000 nm, capable of encapsulating therapeutic molecules by targeting transport processes in the CNS, can improve the crossing of molecules through the BBB in neurological disorders and reach targeted brain regions [29,30,31]. Soft NP, such as nanoliposomes (NL) or exosomes, represent one of the most effective drug delivery processes to be able to protect therapeutic agents and deliver them to the target tissues [32,33]. A growing number of studies report a restorative effect of NL on cellular and animal models of neurological disorders (stroke, Parkinson’s disease, AD) [34,35,36], suggesting improved NP-mediated bioavailability in the CNS.

Furthermore, nanotechnology can be used to improve the bioavailability of PUFA in the form of NL [34,37]. NLs are spherical and closed vesicles with diameters in the nanometer range, formed by phospholipid bilayers dispersed in an aqueous medium [38,39]. NL can be designed to contain n-3 PUFA-rich phospholipids with beneficial neuroprotective properties. Moreover, NL can be used to encapsulate a variety of molecules, including hydrophilic or hydrophobic drugs, proteins, or DNA [40]. They represent an excellent drug delivery process because of their biofilm characteristics which closely resemble those of cell membranes. In addition, NL forms a protective barrier that protects the molecules from degradation by enzymes in the mouth and stomach, digestive juices, bile acid, and intestinal microorganisms [41,42].

In this work, we review the clinical and physiopathology of SAD, including research models of AD, followed by a discussion of current treatments focusing on prevention strategies and the use of soft NP such as NL and exosome as innovative approaches for a preventive strategy in reducing the risk of AD.

## 2. Understanding Clinical Spectrum of AD

### 2.1. Description

#### 2.1.1. Clinical

AD is defined as a neurodegenerative disease that develops over several years, leading to insidious onset cognitive disorders. It is characterized by the appearance of multiple cognitive deficits progressively increasing with time, including memory deterioration in the acquisition of new information and in the recovery of information [13], as well as the association of one or more of the following dysfunctions: aphasia, apraxia, agnosia, or dysexecutive syndrome. AD can be considered an amnestic syndrome of the hippocampal type. These neuropsychological disorders cause impairment in activities of daily living and represent a cognitive and functional decline compared to the previous levels of the individual [16].

Researchers have, in recent years, shown a growing interest in neuropsychiatric symptoms and behavioral disorders such as psychotic symptoms, depression, apathy, aggression, and sleep disturbances [43,44,45]. As a result, in 1996, the concept of behavioral and psychological symptoms of dementia was presented by the International Psychogeriatric Association to designate symptoms of disturbance of perception, the content of thought, mood, and behavior frequently appearing in subjects with ND [46].

AD can be viewed as a process of chemical, physiological and anatomical changes in the brain that can be identified many years before the onset of clinically noticeable cognitive-behavioral syndromes (CBS) [47] (Figure 1).

#### 2.1.2. Pathophysiology

AD results from significant structural and functional damage in the CNS. Two distinct histological lesions have been identified in AD etiology: amyloid plaques and NFT. NFT formation begins in the internal part of the temporal lobe.

The lesions may even be present in these hippocampal structures while the person demonstrates no symptoms of cognitive decline. The NFT then evolves in the external part of the temporal lobe before spreading to the posterior cortical associative areas and the entire cortex [6]. This topography of the lesions corresponds to the evolution of the symptoms of AD [48]. On the other hand, unlike the topography of NFT, the distribution of amyloid deposits is more diffuse. Indeed, these are found first in the neocortex, followed by the hippocampal region, the subcortical nuclei, and finally, in the cerebellum [5].

Amyloid plaques result from the aggregation and abnormal accumulation of the Aβ peptide in the extracellular medium outside the neurons (Figure 1A). Aβ peptide is produced by the amyloidogenic pathway following the sequential proteolytic cleavage of amyloid precursor protein (APP) by β- and γ-secretases [49]. Oligomers of soluble Aβ could interact with the cell surface potentially by direct membrane interaction or by binding a putative receptor leading to impairment of signal-transduction cascades, modified neuronal activities, and release of neurotoxic mediators by microglia, leading to early altered synaptic functions and plasticity [50]. Indeed, the initial oligomerization of soluble Aβ has been found to instigate synapse deterioration, inhibit axonal transport, impact astrocytes and microglia, plasticity dysfunction, oxidative stress, insulin resistance, aberrant tau phosphorylation, cholinergic impairment, selective neuron death [51,52]. Other factors, including abnormal lipid and glucose metabolism, neuroinflammation, cerebrovascular abnormalities [53], and endosomal pathway blockade [54], can also contribute to AD pathology in the brain [55]. The vascular system becomes impaired and fails to deliver sufficient blood and nutrients to the brain and clear away the debris of metabolic products, leading to chronic inflammation by the activation of astrocytes and microglia.

The E4 isoform of the lipid-carrier protein apolipoprotein (apo)E, which is a significant AD risk factor, has been associated with increased Aβ production and impaired Aβ clearance. ApoE4 itself can be cleaved into toxic fragments that affect the cytoskeleton and impair mitochondrial functions [56], which may have a direct consequence on ApoE-mediated clearance of Aβ.

Hyperphosphorylation of Tau proteins (Tubule Associated Unit, Tau) in neurons of the CNS leads to the abnormal conformation of Tau into pairs of helical filaments, which in contrast to amyloid plaques, aggregate inside the neurons to form NFT. Tau proteins detach from microtubules which disrupt intracellular transport, causing dysfunction of neurons and impaired brain activity, which can even lead to macroscopic brain atrophy and death [12].

Interestingly, unlike NFT, no correspondence was observed between the distribution of amyloid deposits and the symptoms of AD patients [48]. The hypothesis of the amyloid cascade suggests that the anomalous accumulation of Aβ peptide and formation of amyloid plaques induce NFT formation and makes neurons more sensitive to neurotoxic effects and neuronal death [57]. Timeline models have been proposed to indicate the process of events carried out during the different stages of SAD [47,58]. For several years, research has shown the role of pathological oligomers in the pathogenesis of AD. This recent understanding of oligomers is supplanting the amyloid cascade hypothesis. While abnormal metabolism of Aβ and Tau proteins are hallmarks of AD and the most trusted identifiers and predictors of AD, a recent paradigm shift has occurred that emphasizes the initial and central role of AβO in AD pathogenesis [11]. Alternative mechanistic models propose that anomalous accumulations of Aβ protein are not necessarily responsible for neurodegeneration. Indeed, amyloid pathology and tauopathy have been shown to appear independently under the influence of genetic and environmental factors [59,60].

Although the sequence of events remains unclear, the presence of both Aβ plaques and NFT processes undoubtedly accelerates AD-related neurodegenerative processes. Initially, before the apparition of Aβ plaques and NFT, the presence of soluble aggregates of Aβ led to the destruction of synapses, dendritic spines and neurons, dysfunction of major system neurotransmitters in the central nervous system, and glutamate and acetylcholine [61]. This includes a progressive loss of cholinergic innervation caused by dendritic, synaptic, and axonal degeneration [53,62].

They cause glutamatergic excitotoxicity and modify glutamatergic synapses and the plasticity process. Cholinergic and glutamatergic systems are important for processing memory, learning, and other aspects of cognition and play a key role in neuronal plasticity. Indeed, glutamatergic and cholinergic deficits are strongly correlated with cognitive deterioration in AD [63]. The cholinergic deficit sets in early histopathologic stage of AD, before the presence of clinical symptoms [64]. Because pathogenic soluble aggregates of Aβ appear early in the sequence of AD, they can represent interesting targets for therapeutics and diagnostics [51].

### 2.2. Diagnostic Criteria for AD

#### 2.2.1. Criteria of the National Institute of Aging and the Alzheimer’s Association (NIA-AA)

In 1984, the diagnostic criteria of the National Institute on Neurological and Communicative Disorders and Stroke and the Alzheimer’s Disease and Related Disorders Association were published. They were based mainly on clinical–pathologic criteria, particularly memory disorders [67]. However, these criteria did not reflect the clinical reality of all patients. Individuals may exhibit evidence of AD biomarkers without cognitive impairment and vice versa. In 2011, new diagnostic criteria of mild cognitive impairment (MCI) and AD were realized by the National Institute of Aging and the Alzheimer’s Association [68,69], with the diagnosis of MCI currently based on clinical, functional, and cognitive examinations.

The most typical MCI associated with AD pathology is MCI-amnestic (single- or multi-domain amnestic-MCI). In amnestic MCI (aMCI), a prodromal stage of AD, there is a memory disorder where the cognitive abilities of the person are inferior in comparison to their age group, gender, and level of education. Nevertheless, this cognitive deficit does not fulfill the dementia criteria. AD can be diagnosed before the onset of dementia if other elements are detected, including amnestic hippocampal syndrome and specific AD biomarkers (Figure 1C).

#### 2.2.2. Specific AD Biomarkers

The NIA-AA and the International Working Group (IWG) consider AD as a slowly progressive neurological disease that begins before the onset of clinical symptoms. Indeed, AD is represented as a continuum that evolves in three stages, asymptomatic (preclinical AD), predementia (MCI due to AD), and dementia (due to AD) [22,70,71].

Although the diagnosis of AD is essentially clinical, the certainty is dependent on evidence of biomarkers indicative of AD-related physiopathological processes. Indeed, the new diagnostic criteria require the use of cerebrospinal fluid (CSF) biomarkers, including total and hyperphosphorylated Tau protein as well as Aβ42 or Aβ42/Aβ40 ratio and positron emission tomography (PET) of τau and amyloid to attribute a probability (high, medium, or low) of the underlying AD-related neurodegenerative processes contributing to the clinical observations [1,22,72]. In 2018, a new AD biological framework and model of pathologic AD biomarkers conceptualized a progressive sequence of neurophysiological, biochemical, and neuroanatomical abnormalities that can be identified years before noticeable CBS.

While not the sole factors, abnormal deposits of protein Aβ and Tau remain hallmarks of AD pathology and make it possible to differentiate AD from other neurodegenerative diseases [71].

Among the pathophysiological markers of AD [58] include the specific presence of amyloid pathology (decreased Aβ1-42 peptide CSF levels or accumulation of the amyloid tracer in PET imaging) and Tau pathology (elevated Tau and phosphorylated Tau protein CSF levels or accumulation of the Tau tracer in PET imaging). Moreover, topographical markers for AD include volume modifications in the brain (temporoparietal, hippocampal atrophy, cortical thickness) assessed by magnetic resonance imaging (MRI) and glucose hypometabolism measured by fluorodeoxyglucose (FDG)-PET [73]. The classification by NIA-AA define and differentiate the Alzheimer’s spectrum of these biomarker criteria using these three amyloid, tau, and neurodegeneration biomarkers.

### 2.3. The Different Stages of the Sporadic form of AD: The Alzheimer’s Spectrum

#### 2.3.1. The Early Asymptomatic Stage: Preclinical Stage

The first neuropathological signs of AD can occur 15 or 20 years before the disease begins [74]. Early Aβ modifications, including oligomerization in the brain, provoke dysfunction in dendrites, axonal processes, and synapses. Nevertheless, the origin of abnormal Aβ and oligomer formation is still unclear [14].

The lesions form slowly without clinical expression (no complaints from patients about troubles in everyday life) [12,14,15]. Preclinical AD is defined as Aβ biomarker evidence of AD pathological changes (PET amyloid retention, low CSF Aβ42) in cognitively healthy individuals or with subtle cognitive changes [71]. Several studies highlight the concept of cognitive reserve in AD, demonstrating that cognition remains stable despite brain Aβ lesions due to compensatory mechanisms (particularly linked with education level) until the early symptomatic stage (MCI). AD can develop at an advanced or very advanced age in people whose cognitive reserve is high.

Indeed, these patients are able to compensate longer for the consequences of AD, thanks to the activation of a more extensive and effective neural network, the cognitive symptoms appearing at a more advanced stage [75,76,77,78]. For the asymptomatic stage, these criteria are used primarily for clinical research protocols rather than for diagnostic purposes.

#### 2.3.2. The Early Symptomatic Stage: Amnesic Mild Cognitive Impairment

In the aMCI, people have cognitive complaints or deficits detected by their entourage. However, there is no significant impact on activities of daily living. An accurate and timely diagnosis of AD is needed, which would allow non-pharmacological and/or drug therapies in the MCI stage or even in the preclinical stage. Many clinical studies are in progress to address this issue [79,80]. In the early symptomatic stage, tests reveal a positive sign of the amyloid and Tau pathology biomarkers [71] without neurodegenerative syndromes. Conversely, the absence of these biomarkers is associated with a low probability. The presence of the two CSF biomarkers, amyloidopathy (low CSF Aβ levels) and neuronal degeneration (CSF Tau and P-Tau levels), are considered indicators of a high risk of conversion to AD [71].

#### 2.3.3. AD

In the typical form of SAD, the patient exhibits all AD symptoms, including a progressive and significant disorder of episodic memory associated with other cognitive disorders (executive functions, apraxia, aphasia, and agnosia).

In many cases, people also have neuropsychiatric disorders such as apathy (49%), depression (42%), aggression (40%), anxiety (39%), and sleep disorders (39%) [81]. These disorders have a significant impact on autonomy requiring external aid to perform the acts of everyday life. At the stage of dementia, the diagnosis is made based on clinical behavioral tests, where biomarkers are used only to increase the threshold of certainty of the diagnosis for atypical forms or young subjects [22].

The presence of biomarkers can be used to indicate the severity of the AD [7,71]: decreased CSF Aβ levels, increased CSF Tau and/or P-Tau levels, cortical thinning and hippocampal atrophy based on MRI, hypometabolism or hypoperfusion of posterior cingulate and temporoparietal cortex (FDG-PET), and detection of cortical amyloid fixation (PET), adding to neurodegeneration syndromes [71,82]. Finally, the certainty of AD diagnosis is evaluated by a level of probability, with definitive evidence provided only by biopsy or autopsy.

### 2.4. Risk Factors of SAD

Based on current research, AD etiology is multifactorial genetic and environmental risk factors that can be categorized as modifiable and non-modifiable factors [83,84,85,86,87].

#### 2.4.1. Non-Modifiable Risk Factors

Several studies suggest that the greatest non-modifiable risk factors for SAD are age, the APOE-ε4 allele, and gender [4,88].

##### Age

The main risk factor for SAD is age. Indeed, increased life expectancy is correlated with a higher probability of developing neurodegenerative diseases, including AD [89,90]. In normal aging, the structure of the brain changes in membrane fluidity and lipid composition, regional brain volume, density, cortical thickness, and microstructure of the white and grey matter. There is a progressive loss of neuronal synapses, leading to a neuronal density decrease.

##### Genetic Risk Factor

While ADAD is caused by the mutation of one of the genes involved in amyloid metabolism, including amyloid precursor protein (APP), presenilin 1 (PSEN1), or presenilin 2 (PSEN2), the main genetic risk factor in SAD is the APOE gene [91,92,93,94]. APOE is associated with the transport of lipids, including cholesterol, in peripheral tissues and in the central nervous system.

By virtue of its role in astrocyte-derived cholesterol transport to neurons, it ensures lipid delivery to neurons and, thus, membrane homeostasis, a critical process in neuron and brain lesion repair. The APOE gene has three alleles: ε2, ε3, and ε4. The ε4 allele is a genetic risk factor of SAD involved with a high risk of AD, found to be associated with atrophic hippocampal volume, abnormal accumulation of Aβ protein and increased amyloid deposits, and cerebral hypometabolism [95].

The ε4 has been linked to changes in neurotoxic and neuroprotective mechanisms, including Aβ peptide metabolism, aggregation, toxicity, tauopathy, synaptic plasticity, lipid transport, vascular integrity, and neuroinflammation [96]. It has been shown that having the ε4 allele increases four-fold the risk of developing AD, whereas the ε2 allele decreases AD risk; the ε3 allele has no effect on AD risk. However, some ε4 allele carriers never manifest AD, which indicates that other as yet to be identified determinants (genetic or otherwise) may be involved in AD development [88].

##### Gender

AD disease prevalence and symptom progression are disproportionately higher in women [97,98,99]. Moreover, there are different risk factors for women and men (APOE genotype, cardiovascular disease, depression, hormonal depletion, sociocultural factors, sex-specific risk factors for women, and performance in verbal memory), which may contribute to this difference. To understand the influence of gender differences, studies are needed to determine gender influence on biomarker evolution across the life span, including cognitive abilities, neuroimaging, CSF, and blood-based biomarkers of AD, particularly at earlier ages [98]. In addition, preclinical and clinical studies in the development of AD therapeutics for both genders are needed.

#### 2.4.2. Modifiable Risk Factors

Modifiable risk factors are of considerable interest since they are a lever for the action of preventive strategies. Cardiovascular damage is a risk factor for neurodegenerative diseases, and since the brain is supplied by a large network of blood vessels, a healthy cardiovascular system can be considered neuroprotective [100]. This could explain why part of the risk factors for cardiovascular disease are in common with AD, including hypertension, dyslipidemia, diabetes, obesity, dietary factors, smoking, and physical activity (Figure 1B). Thus, lifestyle is also an important risk factor, where intellectual, physical, and social activities, as well as diet, may help to prevent AD [1,22,100,101,102,103].

##### Metabolic Disorders and Dyslipidemia

Although the brain represents 2% of the total body weight, 20% of body oxygen consumption and 25% of the glucose consumption can be attributed to this organ [101,104]. The brain is the most lipid-rich organ next to the adipose tissue. Indeed, lipids are part of gray matter, white matter, and nerve nuclei and are needed for neuronal growth and synaptogenesis. Lipids in the brain are composed of 50% phospholipids, 40% glycolipids, 10% cholesterol, cholesterol esters, and trace amounts of triglycerides [105].

Long-chain polyunsaturated fatty acids (LC-PUFAs) represent 25–30% of the total fatty acids (FAs) in CNS, in particular, docosahexaenoic acid (DHA) and arachidonic acid (AA). Cholesterol and long-chain omega-3 FAs, and especially DHA, play major roles in brain function. Research shows that the imbalance of lipid homeostasis is associated with a high risk of AD [105,106]. The brain is the organ that is the richest in cholesterol, containing 25% of all cholesterol in the human body. The cholesterol used in the brain is synthesized within the CNS. Altered cerebral cholesterol homeostasis may promote neurite pathology, Tau hyperphosphorylation, and the amyloidogenic pathway [49].

Increased brain cholesterol levels and dyslipidemias overall have been linked to AD incidence [49,55,100,107]. It could be assumed that similar to the etiology of cardiovascular disease, diabetes, and obesity, these changes in lipid homeostasis can increase the risk of age-related neurodegeneration and AD with time. Dyslipidemias are also associated with obesity, which has been linked to insulin resistance/hyperinsulinemia in the development of AD [108,109]. Indeed, brain insulin resistance has been shown to be involved in cellular and molecular mechanisms of neurofibrillary tangles formation and amyloid plaques [110], leading to AD being referred to as type III diabetes. These studies clearly indicate that preventive strategies aimed toward maintaining optimal brain lipid status may be useful in maintaining neuronal functions and synaptic plasticity, thereby reducing AD risk. Lipids provide a further advantage in that dietary intervention represents a fairly straightforward manner to achieve proper lipid homeostasis [111,112,113].

##### Other Risk Factors

Low levels of cognitive, social, and physical activity may be linked to a greater risk of developing neurodegenerative diseases [3,83]. An enriched environment appears to favor the establishment of a cognitive reserve that includes the level of education (level of study, profession), the quality of social interactions, the variety of leisure activities, and the practice of physical exercise. Nevertheless, these cognitive and physical factors are not alone in affecting the reserve capacity. Other factors are involved, including nutrition and other environmental parameters that can protect cardiovascular pathologies, thereby reducing AD risk [87]. Symptoms of depression, anxiety, stress, and chronic psychological distress have also been associated with an increased risk of MCI and AD [114,115]. Moreover, excessive consumption of tobacco and alcohol increases cognitive impairments [20]. A history of head trauma and hearing loss may favor the onset of AD [3,102,116]. Recently studies have shown that air pollution may be linked to an increased risk of neurodegenerative diseases [117].

A recent report highlighted 12 modifiable risk factors representing about 40% of dementias in the world: a low level of education, hypertension, hearing impairment, smoking, obesity, depression, sedentary lifestyle, diabetes and poor social contact, excessive alcohol consumption, history of head trauma and air pollution [117]. These last three factors have been updated recently [3].

## 3. Research Models Used for AD

In order to both study the underlying mechanisms and to test and identify possible preventative strategies, AD models [118] range from in vitro cell culture models (two-dimensional (2D) or three-dimensional (3D)) to animal models. Generally, 2D cell culture models lack cues provided by the extracellular matrix (ECM). These models are easy to maintain and cost-effective but do not allow the study of glia-neuron communication and crosstalk [119]. 3D models include neurospheroids [120], organ-on-a-chip devices [121], and engineered brain tissue [122], mimicking the brain’s complexity; however, they are more difficult to engineer and maintain [123]. Animal models are required for studying both physiological and behavioral mechanisms of AD, but interspecies differences may result in unexpected results in clinical trials [123,124]. The various AD models presented here are summarized in Table 1.

### 3.1. 2D In Vitro Models of Alzheimer’s Disease

Researchers have attempted to simulate or induce the clinically observed increased Aβ42/40 ratio to study AD in vitro. Among the 2D models of interest, the most common human cell types used include human embryonal stem cells (hESC), induced pluripotent stem cells (iPSC) from AD patients, or neurons from AD patients with relevant mutations [118]. For example, Koch et al. observed an increased Aβ42/40 ratio due to a decrease in Aβ40 hESC-derived neurons overexpressing PSEN1 [125], also found by Mertens et al. [126]. Increased Aβ42 was only found in neurons derived from patients with a specific APP K724N mutation. Neurons are not alone in playing an important role in the Aβ42/40 ratio, as shown by Liao et al., who demonstrated secretion of Aβ42 by hiPSC-derived astrocytes [127]. Oksanen et al. used PSEN1 ΔE9 mutated hiPSC-derived astrocytes to show an increased Aβ42/40 ratio, as well as increased reactive oxygen species (ROS) and increased cytokine release [128], suggesting an increase in the pro-inflammatory response. Jones et al. used the PSEN1 M146L mutation to show disturbed astrocyte marker expression [129]. PSEN1 is not only responsible for the increased Aβ42/40 ratio but is also involved in mitochondrial impairment. Martin-Maestro et al. used hiPSC-derived neurons with a PSEN1 A246E mutation to show the role of mitochondrial dysfunction in AD [130].

Interestingly, Perez et al. used human iPSC with a loss of PITRM1 function (Pitrilysin metallopeptidase 1), an enzyme involved in mitochondrial degradation associated with AD, as a 2D model and to form 3D organoids [146]. They showed that only the organoids were able to provide the increased Aβ42/40 ratio and higher p-Tau levels, suggesting the need for cell-cell and cell-matrix interactions to fully simulate AD in vitro.

Lastly, primary murine neurons and cell lines are often used to model AD. The ReN immortalized neural stem cell line contains various APP mutations and can differentiate toward neurons or glia cells, rendering it an ideal cell line for AD modeling [131].

Additionally, PC12 is an immortalized clonal cell line showing GLP-1 neuroprotection and Aβ plaque formation [132]. However, 2D models lack the proper cellular environment and support cells to model AD properly; hence, 3D culture models provide a solution.

### 3.2. 3D Models of Alzheimer’s Disease

3D models of AD offer the complexity of the brain without the ethical constraints of animal models [147]. Organoids, or other forms of 3D cell culture, can simulate AD pathology, as they are able to secrete sufficient levels of Aβ42 to form Aβ plaques and form NFT, unlike 2D cultures [148]. Hernandez-Sapiens et al. have used iPSC-derived neurons with the PSEN1 A246E mutation, as seen before, to simulate AD in vitro [133]. They were able to generate Aβ oligomers, representative of AD in vivo, by culturing the cells on a Matrigel platform. Ranjan et al. formulated poly(lactic-co-glycolic acid) (PLGA) scaffolds using wet electrospinning to encapsulate iPSC-derived neural progenitor cells (NPCs) and mimic the brain structure (Figure 2A) [134]. They were able to acquire pathogenic levels of Aβ42 and p-Tau using AD patient-derived iPSCs. Papadimitriou et al. employed a starPEG heparin-based hydrogel, which could incorporate both neural stem cells and pathogenic levels of Aβ42 to simulate AD-like physiopathology [135]. They demonstrated that Aβ42 was responsible for the loss of neural plasticity, similar to that observed in AD (Figure 2B).

Therefore, this system could allow for the identification of therapeutic targets to reduce the loss of neural plasticity. Moreover, Cairns et al. induced AD in human-induced neural stem cells in a silk protein scaffold using the herpes simplex virus type I (HSV-1) [136]. They were able to mimic the plaque formation, neuroinflammation, and decreased functionality of the cells without the use of exogenous AD mediators. It should be noted that after plaque formation, the model can no longer be used to evaluate preventative strategies.

Other researchers have assembled spheroids to obtain a 3D environment. For example, Lee et al. used iPSCs from various AD patients’ blood to form Aβ oligomers in neuro-spheroids, which provide a platform for high throughput testing of AD drugs [137]. Human NPCs were used to form differentiated 3D neurospheroids with Aβ aggregation, modeling AD [149]. Kwak et al. continued this research and assembled 3D spheroids expressing different levels and ratios of Aβ42/40 (Figure 2C) [138]. They were able to show p-Tau pathology and the importance of lowering the Aβ42/40 ratio to reduce AD-related neurodegenerative processes.

The use of microfluidics to create neurospheroids or brain-on-a-chip devices has been extensively used by researchers as well. For example, Cai et al. used an acoustofluidic platform to aid in the high-throughput formation of homogeneous neurospheroids [139]. Acoustic soundwaves allow for the rapid formation of spheroids, and the addition of Aβ aggregates into the spheroids resembles AD pathology (Figure 2D).

Moreover, Park et al. used a microfluidic system for a 3D triculture of astrocytes, neurons, and microglia [140]. The model showed Aβ aggregation, p-tau accumulation, and secretion of inflammatory cytokines can be used to study the pathogenesis of AD.

### 3.3. In Vivo Animal Models of AD

The genetic mutations found in ADAD and risk factors associated with SAD have provided useful information to develop animal AD models, particularly in mice [150]. This allows not only targeted genetic modifications related to AD pathology but also the evaluation of associated cognitive deficits [151]. Many transgenic mouse models are based on APP mutations, which can also be used in 2D in vitro cell-based models. However, the results have not been entirely conclusive.

Chen et al. demonstrated the relationship between a type of cognitive performance and b-amyloid plaque deposition. APP overexpression in mice showed increased Aβ plaque formation, as well as a spatial learning decline, but not all forms of learning and memory [141].

More recently, Ochiishi et al. used Aβ tagged with Green Fluorescent Protein (GFP) in mice to observe the cascade of events leading from Aβ oligomeric formation to synaptic dysfunction in vivo [142]. These mice showed elevated levels of p-Tau, impaired recognition memory, and altered spine morphology.

Additionally, injection mice models are often used to model AD in vivo. Injection in tauP301L transgenic mice resulted in elevated p-tau levels close to the site of injection [143]. Similarly, T40PL-GFP transgenic mice, with the P301L 2N4R tau mutation, induced p-tau pathology after three months [144]. Moreover, intracerebroventricular (ICV) administration of Aβ oligomers into mice shows a failure to induce glucose intolerance, indicating Aβ oligomers target metabolic control [152]. Similarly, intra-hippocampal injections of Aβ oligomers showed loss of memory, correlated with the ERK1/2 pathway activation, which is involved in memory function [145]. The interest in these injection models is that they are not genetic forms of AD and are more similar to the SAD model. By exposing these models to certain risk factors, we get closer to a model more representative of human pathology and epidemiological data.

Researchers have been able to simulate AD lesions, including amyloid plaque formation and Tau pathology, in various models, leading to the identification of therapeutic targets as well as drug testing. However, understanding the molecular mechanisms involved in the early onset of AD will help to develop strategies to reduce AD risk or even prevent the disease altogether. More research is required in order to develop an adequate model of the very early stages of AD, especially in models with sporadic forms, which represent the majority of AD cases.

Non-human primates (NHP) can be used as models of sporadic age-associated brain β-amyloid deposition and pathological changes in AD. Recently, Latimer et al. showed that vervets and other NHP are promising models for exploring early-stage disease mechanisms and biomarkers and testing new therapeutic strategies [153]. These monkeys have the propensity to develop diseases relevant to humans during aging without genetic handling [154]. However, there are several limitations; vervets show amyloid deposits but do not have neurofibrillary tangles or tauopathy; therefore, they do not present generalized neurodegeneration. Vervets are best considered a pattern of early amyloid pathology and corresponding behavioral and biomarker changes, making them important for the study of the early stages of AD [153].

## 4. Current Treatments and Prevention Strategies for Alzheimer’s Disease

### 4.1. Drug Treatments

#### Description

Current drug treatments for AD are symptomatic-based rather than curative to limit the progression of cognitive symptoms and behavioral and psychological symptoms of dementia (BPSD). Four drugs (donepezil, memantine, galantamine, rivastigmine) are approved on the market and belong to two families: anticholinesterase inhibitors and anti-glutaminergics. These treatments are delivered through the oral or transdermal route [155,156]. Anticholinesterase inhibitors are molecules designed to increase acetylcholine levels in the brain, which is a molecule that allows the transmission of information between certain neurons and plays a role in memory. These treatments are intended to correct the acetylcholine deficiency that is observed in the CNS of persons with AD. Anti-glutaminergics are used to regulate glutamate levels through a noncompetitive antagonist effect of N-methyl-D-aspartate (NMDA) receptors. Glutamate is a neurotransmitter that has a role in the brain functions of learning and memorization. High levels of glutamate are likely to cause pathological effects causing the death of neurons. These drug treatments are used in order to delay the evolution of the disease, to stabilize or to improve, albeit temporarily, the cognitive functions, and to control the disorders of the behavior. Although not curative, these treatments nevertheless help to maintain independence and improve the quality of life for AD people and for their caregivers. However, these treatments, whose effectiveness is only partial and temporary at best, affect only the consequences of AD rather than the cause [1,17,157,158]. These drug therapies may be more beneficial in the early asymptomatic stage before the process of neurodegeneration occurs. Other reasons also contribute to the modest effectiveness of these treatments, including the difficulty in brain drug targeting due to restricted passage from the circulation to the CNS through the BBB [159]. Indeed, many drug trials fail because of permeability issues at the BBB in AD.

Because of this, the increased dosage is necessary, which could also increase the possibility of secondary undesired effects [160,161]. The BBB represents a challenge for CNS drug delivery, and many strategies have been developed to address this challenge [162]. Drug efficacy may also be reduced due to age-related modifications in neuronal membranes and membrane receptors, which is not necessarily considered in pre-clinical studies. Indeed, a recent study showed changes in the microdomains of synaptosomes isolated from aged mice, which increased their response to amyloid stress and inhibited the neuroprotective effects of the ciliary neurotrophic factor [163].

Another limitation in treatments may be due to their administration during the late stages of AD. For example, studies of mice with genetic mutations of ADAD causing early and rapid accumulation of amyloid plaques have allowed testing of anti-amyloid immunization to remove amyloid plaques [164]. However, in numerous human clinical trials using this approach, there was a decrease in amyloid load, but without significant clinical improvement nor reduction in disease progression [165]. It is, therefore, possible that the treatments are administered at a time when AD is already in the advanced stages and are thus less effective. Timely intervention may be important and emphasizes the need for better diagnosis of the early stages of AD using additional biomarkers [158]. Indeed, rapid and accurate diagnosis should take into account target populations with risk factors, including family history (genetic factors including the ε4 allele) and isolated memory complaints. Drug development has targeted amyloid plaque formation, but other novel targets need to be explored [155].

The absence of effective curative treatments and difficulties in accurately diagnosing early-stage AD clearly demonstrates the need for implementing preventive and neuroprotective strategies to slow down the neurodegenerative process (neuronal dysfunctions: axons, dendrites, synapses), thereby reducing AD risk [166].

### 4.2. Non-Pharmacological Therapies

Non-pharmacological therapies, in addition to drug treatments, represent an alternative to the treatment of neurodegenerative diseases [156]. Several studies and international trials are completed or progress to investigate the multidomain intervention in AD [166,167,168,169,170,171,172], which is an approach using multiple activities. Indeed, research shows a positive correlation between increased physical activity, cognitive training, improved nutrition, and slowing cognitive and functional decline and intensity of BPSD. Nevertheless, these activities have been carried out over short periods, with little information on long-term studies.

### 4.3. Prevention Strategies for Alzheimer’s Disease

A review of the current literature highlights the interest in prevention and non-drug therapies, specifically for MCI or the preclinical stage. As discussed above, AD abnormalities such as Abeta-induced synaptic dysfunctions or endosomal pathway blockade occur early, progressing with time, even over decades, by which time the available treatments have only modest effects. The decreased cognitive performance begins around midlife from 45 years [173] and increases with age due to structural and functional changes in the brain (for instance, regional brain volumes and integrity of the white matter, reduction in the fluidity of brain membranes, and changes in lipid composition) [163,174,175,176]. Identification of modifiable risk factors is, therefore, essential in order to define preventive actions against this insidious disease. Here, because of the importance of lipids in brain structure and function and the relative ease with which lipid status can be optimized by dietary intervention, we focus on this as a modifiable risk factor to target using diet as a preventive strategy for reducing age-related cognitive impairment and risk of preclinical stage of AD.

#### 4.3.1. Dietary Intervention

##### Fatty Acids: Omega 3, 6

PUFA has a key role in the production and storage of energy, synthesis and fluidity of cell membranes, and enzymatic activities, among others [23]. Two specific PUFAs, linoleic (LA, C18:2n-6) and alpha linolenic (ALA, C18:3n-3) acids, are essential since the body is unable to synthesize these fatty acids, and the only way to obtain them is from dietary sources [177]. LA and ALA are precursors for arachidonic acid (AA, C20:4n6), eicosapentaenoic acid (EPA, C20:5n-3) and DHA (C22:6n-3). PUFA is required for brain development, integrity, and function. Omega-3 (n-3) and 6 (n-6) are important components of biomembranes and have a key role in the integrity, development, neuron maintenance, and functions, including synaptic processes, neuronal differentiation, and neuronal growth [178,179,180].

##### Docosahexaenoic Acid

The brain has a high level of n-3 fatty acid, DHA, mainly in membrane photoreceptors and synaptic membranes. It is present particularly in membrane phospholipids (PL) of phosphatidylethanolamine (PE) and phosphatidylserine (PS), with smaller amounts also found in phosphatidylcholine (PC). DHA is known for its neuroprotective effects and synaptic plasticity and has a key role in aging, memory, vision, and corneal nerve regeneration [181]. DHA accounts for more than 90% of n-3 PUFAs in the brain and 10% to 20% of total lipids and is found in particularly high levels of gray matter. The total volume of gray matter decreases with age, which corresponds to a loss in the composition of DHA [182]. DHA can influence cellular and physiological processes, including membrane fluidity, the release of neurotransmitters, myelination, neuroinflammation, and neuronal differentiation and growth [183]. Long-chain n-3 FA supplementation in the early stages of AD has shown promising potential [111]. Studies show a relationship between a diet rich in fish-derived n-3 FA and cognitive performance during aging [184].

Fish oil is an excellent source of DHA and has been studied as a potential preventing food supplement for AD. However, it seems that only patients with mild cognitive impairment who do not have ε4 allele of the APOE gene had better cognitive outcomes after treatment with fish oil [185]. This would suggest the importance of n-3 FA supplementation before the onset of AD symptoms, especially in people at risk. In fact, DHA is able to attenuate molecular mechanisms that are deleterious to the CNS in the early stages of AD. The modification in the fluidity of neuronal membranes is involved with brain aging and may play an important role in AD [106]. DHA supplementation in the diet could prevent age-related neuronal membrane changes and associated impairments. Furthermore, DHA supplementation can support reactivity to molecular therapeutic targets impaired in AD, such as the ciliary neurotrophic factor (CNTF), suggesting that DHA may be of more value in combination with other treatments, such as neuroprotective molecules, than alone [163]. Various in vitro and in vivo studies have described that DHA can have neuroprotective effects against neurotoxicity induced by Aβ [186]. DHA has been shown to lower Aβ production in various AD models [187], improve blood flow and decrease inflammation [188], further demonstrating its potential to reduce AD risk.

Currently, several clinical trials are testing the neuroprotective effects of n-3 PUFA administration in patients with AD. Despite these numerous studies, the results in AD patients have not been consistent in terms of cognitive or neuronal improvement. In addition, the effects observed are generally short-term. These discrepancies can perhaps be explained by numerous factors, including the administration of DHA only in late-stage AD, insufficient age-influenced bioavailability, or the formulation of DHA used.

In the AD, delivery of bioactive substances to the CNS is a very complex process and the development of prevention and therapeutic strategies is challenging. Research is being carried out to test new strategies to improve the penetration of molecules into the CNS.

## 5. BBB and Administration Strategies for Alzheimer’s Disease Treatment

### 5.1. BBB and Soft Nanoparticles

The BBB’s main role is to separate the blood that circulates in the body, and that might contain toxic foreign molecules from the brain’s extracellular fluid present in the CNS [189]. The German bacteriologist Paul Ehrlich discovered the BBB in 1885 when he successfully stained all animal organs except the brain following injection of aniline dye solution into the peripheral circulation. This discovery was later confirmed by his student Edwin Goldman in 1913 [190]. However, only once scanning electron microscopy (SEM) was invented in 1937 could the actual BBB membrane be observed. Anatomically, the BBB’s structure includes the structure pericytes, astrocytes, neurons, and microglia (Figure 3) [191]. Due to the presence of high electrical resistance and tight junctions between endothelial cells caused by their encapsulation by the pericytes and astrocytes, only water and other small molecules can penetrate without restrictions by passive transcellular diffusion through the BBB [189,192]. Conversely, molecules such as drugs, amino acids, and glucose are polar, hydrophilic, or possess a high electric charge to cross the BBB through active transport routes that depend on specific proteins [193]. Therefore, due to the rise in neurodegenerative diseases, the challenge of delivering and releasing drugs or bioactive compounds into the brain is attracting much attention. Soft nanoparticles, such as liposomes and exosomes, are nanovesicles that are able to deliver drugs and genes across the BBB (Figure 3) [33,37].

#### 5.1.1. Liposomes

In the 1960s, Dr. Bangham discovered liposomes when he noticed that phospholipids, when surrounded by an aqueous medium, formed a closed bilayer [194,195]. Being amphiphilic, the hydrophobic acyl chains of phospholipids lead to the thermodynamically favorable formation of lipid spheres upon contact with water [42,196]. Electrostatic interactions, such as van der Waals forces and hydrogen bonding, further enhance this formation [197,198]. Liposomes generally consist of a lipid bilayer encircling an aqueous core [199,200]. Liposomes are able to encapsulate hydrophobic drugs in their bilayer and hydrophilic bioactive molecules in their core [201]. With the lower volume of hydration in the liposome core, hydrophilic molecules are entrapped with a lower efficiency than hydrophobic molecules [42].

Liposomes can be classified as conventional, PEGylated/stealth, or ligand-targeted based on the characteristics of their surface [202]. Liposomes have been researched for more than five decades, to the point where they are well-established drug delivery vectors, resulting in the marketing authorization of several clinically approved liposomal-based products [203]. Indeed, they offer better biocompatibility and safety than polymeric and metal-based nanoparticles due to their resemblance to biomembranes [204,205].

AD drugs fail to generate therapeutic effects in part because they cannot pass through the BBB to enter the CNS. Even though conventional liposomes cannot cross the BBB, modifying their surface can enable them to pass through and unload their cargo directly into the CNS [206]. Proteins, peptides, and antibodies receptors found on the BBB’s surface can be used to mediate the translocation of liposomes via receptor-mediated transcytosis.

Transferrin (Tf)-functionalized liposomes have been used for BBB targeting since a transmembrane glycoprotein overexpressed on brain endothelial cells called the transferrin receptor (TfR) is one of the most commonly targeted receptors. The problem of endogenous Tf binding inhibition to the TfR is usually resolved by avoiding ligand competition and using specific antibodies against TfR [207,208,209,210]. Likewise, the mammalian cationic iron-binding glycoprotein lactoferrin (Lf) is overexpressed on the BBB. Lf-modified liposomes have also been created to cross the BBB via receptor-mediated transcytosis [211,212].

Simultaneously, cationic liposomes can penetrate the BBB via absorption-mediated transcytosis, taking advantage of the BBB’s negative charge and consequently triggering, through electrostatic interactions, the cell internalization processes [213,214,215]. However, binding to serum proteins and the nonspecific uptake of cationic liposomes by peripheral tissues are major drawbacks that require the administration of highly toxic doses of liposomes to achieve therapeutic efficacy [32].

Another strategy is to bind nutrients, such as glucose and glutathione, to the surface of liposome to facilitate its translocation via carrier-mediated transcytosis [216]. Nutrients are normally transported to the brain from the blood by selective transporters overexpressed on the BBB’s surface, such as amino acid, hexose, or monocarboxylate transporters. For this, glucose-functionalized liposomes have been developed to improve their transcytosis through the BBB [217,218]. G-Technology^®^ targets liposomes through the BBB by modifying them using glutathione that targets the glutathione transporters highly expressed on the BBB’s surface [209,219,220].

Moreover, developing liposomes with more than one targeting ligand has been successfully used as a new strategy to deliver therapeutics to the brain. These bifunctional liposomal delivery carriers increase BBB targeting efficiency, most likely by overcoming the receptor or transporter saturation limitation of monofunctional liposomes [221]. Small molecules that present high affinity towards amyloid peptides were successfully loaded in bifunctional liposomes to create enhanced multifunctional carriers [208,209,210,222,223,224]. With regards to regulatory aspects, liposomes are one of the most popular nanocarrier systems available for the loading and delivery of drugs and genes, as can be seen from the increasing numbers of Investigational New Drug (IND) application submissions [32].

#### 5.1.2. Exosomes

Many similarities exist between exosomes and liposomes, as both of them range in size from 40 to 120 nm and are composed of a lipid bilayer. Nevertheless, they have major differences as well, such as the exosomes’ complex surface composition. The unique lipid composition of exosomes differentiates them from other nanovesicles and dictates their in vivo fate due to their role in interactions with serum proteins. Tetraspanins and other membrane proteins increase the efficiency of exosome’s targeting ability by facilitating their cellular uptake.

Moreover, exosomes are more biocompatible, can evade phagocytosis, and have an extended blood half-life compared to liposomes, micelles, and other synthetic soft nanovesicles [225,226,227]. Exosomes’ smart targeting behavior of specific receptors is governed by the donor cells in the form of lipid and cellular adhesion molecules found on their surfaces [228]. Furthermore, their high biocompatibility and low immunogenicity enhance their uptake profile, their stability in systemic circulation, and their in vitro and in vivo therapeutic efficacy [229,230].

The exosomal content depends on the originating cell or organism, but in general, all exosomes contain non-coding RNAs, microRNAs, mRNAs, small molecule metabolites, proteins, and lipids [231]. In addition, their surface contains valuable receptors responsible for the identification of exosomes and the transportation of encapsulated materials to recipient cells [232]. Exosomes can be isolated using filtration, polymer-based precipitation, chromatography, differential centrifugation, ultracentrifugation, and immunological separation. Research is actively ongoing with the goal of developing a gold-standard universal method that is efficient for isolating exosomes (with a high yield) and does not compromise their biological function [233].

Previously, exosomes and their cargo have been shown to have a central role in normal CNS communications, immune response, synaptic function, plasticity, nerve regeneration, and the propagation of neurodegenerative diseases [232]. The important role of exosomes in normal and diseased brain states indicates that they may also play a significant role in the pathogenesis of mental disorders [234].

One important finding in biomarker and drug delivery research was that the content of exosomes can remain active after they cross the BBB. The effective brain delivery of siRNA-loaded exosomes by systemic injection in mice has been successfully demonstrated by Alvarez-Erviti et al. [235].

To reduce immunogenicity and to target exosomes to the brain, they isolated exosomes from self-derived dendritic cells, targeted them with lysosome-associated membrane protein 2 (Lamp2b) fused to the neuron-specific rabies virus glycoprotein (RVG) peptide, and loaded them with siRNA via electroporation. These engineered targeted exosomes caused a specific gene knockdown exclusively in the brain following the delivery of GAPDH siRNA. Other researchers used intranasal injections to successfully deliver exosomes to the brain of mice [236]. Recently, evidence of brain–body communication via exosomes was identified when Gómez-Molina et al. recovered exosomes expressing a fluorescently tagged protein that is only found in the brain from the blood of rats [237].

Even though the exact mechanism of exosome transport across the BBB is not fully characterized, these studies prove that exosomes in circulation can access the brain and vice versa. The transport of exosomes across the BBB has been hypothesized to be through [33]: (1) nonspecific/lipid raft; (2) macropinocytosis; (3) cell signaling induced by the adhesion and fusion of exosomes to the cell surface, which causes the release of the exosomal content; (4) a signaling cascade induced by the association of exosomes with a protein G-coupled receptor found on the cell surface; or (5) receptor-mediated transcytosis.

Exosomes’ use as drug-delivery vesicles to the CNS has become a topic of interest since their competence to penetrate the BBB was discovered, and they have already been used as drug-delivery nanocarriers in cancer and neurodegenerative diseases. Yang and coworkers reported that the delivery of anticancer drugs encapsulated in exosomes across the BBB significantly decreased the fluorescent intensity of xenotransplanted cancer cells and tumor growth markers [238]. A formulation of catalase loaded in exosomes accumulated in Parkinson’s disease mouse brain and provided significant neuroprotective [239]. Liu et al. treated morphine addiction using exosomes expressing RVG peptide loaded with opioid receptor mu (MOR) siRNA [240].

The siRNA-loaded exosomes strongly inhibited the morphine relapse after delivering efficiently and specifically the MOR siRNA into the mouse brain, which significantly reduced the MOR mRNA and protein levels.

These studies showcase the promising role of nanoliposomes and exosomes as smart drug delivery systems able to cross the BBB and target brain tissues. The different drug administration techniques are presented below.

### 5.2. Oral Administration

Administration of therapeutic molecules by the oral route or prevention strategies, including dietary intervention, are the simplest to apply and are used both in clinical trials with patients and in studies using animal models. It nevertheless has drawbacks due to limited passage through the intestinal barrier and the BBB, as well as clearance by other organs, including the liver and kidney [241]. Nanocarriers, including nanoliposomes or exosomes, can be used to overcome these different obstacles and to enhance drug bioavailability and pharmacokinetic behavior [242,243], as discussed above.

### 5.3. Intravenous and Intracerebral Administration

Therapeutics can be injected directly into the circulation (intravenous) to overcome the intestinal barrier, but the BBB remains an obstacle. Other methods of administration to avoid the BBB consist of directly injecting the molecule, either intraventricularly or intrathecally, into CSF or into brain parenchyma. Intracerebral or intracerebroventricular injections using stereotaxy are a common procedure for in vivo animal models. In humans, this is invasive and costly since it requires surgical intervention and can be painful (intrathecal injections) [241,244]. Because of this, these options are only considered in severe conditions, when the patient is most likely already hospitalized. Nevertheless, this approach is of interest for introducing slow-release implants or a colony of stem cells in the CNS for the timed release of therapeutic molecules [245].

### 5.4. Intranasal Administration

Intranasal (IN) administration provides an attractive and promising alternative for drug delivery to the CNS. It is non-invasive, painless, non-stressful, and relatively easy to perform without requiring a medical specialist [246,247,248]. The IN route bypasses the BBB, enhancing drug bioavailability by avoiding first-pass metabolism and intestinal degradation. Drugs can be directly administrated through the IN route to the CNS via the olfactory mucosa [249]. The small volumes that can be applied intranasal is a limiting factor and, as such, requires concentrated solutions. Nevertheless, nasal mucosa provides a large surface area and rich vasculature for very efficient drug absorption. Recent studies have analyzed the direct transport of proteins and peptides via IN [250]. Currently, no nanosystem has passed the clinical development stage. There are several in vivo studies, but conclusions are not yet clear due to differences in nasal anatomy between humans and animals. This approach is increasingly used in clinical studies for the treatment of neurological diseases [251] like AD [32,252], brain injuries [253], and autism [254]. Currently, more than a hundred clinical trials are ongoing and investigating intranasal drug delivery, especially in CNS-related diseases, with the greatest promise appearing to be IN the administration of insulin or oxytocin. Even if the intranasal mechanisms of drugs to the brain are not yet completely understood, IN delivery represents a promising pathway of administration and should be investigated for future pre-clinical and clinical studies in the treatment of neurological diseases such as AD [255].

### 5.5. Novel Administration Strategies

Technological advantages have allowed the emergence of novel strategies in recent years to potentiate the delivery of therapeutic substances.

#### 5.5.1. Ultrasound and Electromagnetism

Several studies have investigated the effects of ultrasound on BBB permeability. Ultrasound can be useful to temporarily open the BBB without damaging healthy tissues. The tight junctions of the BBB can be opened reversibly by focused ultrasound-induced microbubble oscillation. Lin et al. used focused ultrasound to reversibly open the BBB to deliver cationic liposomes loaded with doxorubicin to optimize glioma targeting capabilities [256].

A study has shown in people with brain tumors that the targeted diffusion of ultrasound can improve the porosity of the BBB and the transport of molecules to the targeted areas. Disruption of the lipid organization of membranes by ultrasound increases membrane permeability and allows greater penetration of the therapeutic molecules [257]. Other research shows that the application of a resonant magnetic field gradient (RMFG) allows the detection of nanometer movements associated with ultrasound.

A magnetic resonance guide can be a good solution to focus ultrasound in combination with intravenously injected microbubbles in order to transiently open the BBB and reduce Aβ and τau pathology in animal models of AD [258]. This method is presented as a safe, reversible, repeatable, and non-invasive method for access across the BBB, and thus, it is a promising method for patients with AD [259,260,261].

#### 5.5.2. Transdermal Delivery Systems via Microneedles

One of the emerging minimally invasive drug delivery tools is microneedles [262,263,264]. Microneedles are capable of delivering therapeutics and nanocarriers in an active or passive fashion. Such tools can be paired with smart wearable electronics to control the dosing of drugs and ensure patient compliance. However, substantial research is needed to ensure the effectiveness of therapeutics delivered subcutaneously. This technology is promising and can be adapted to a range of drug-delivery applications [265]. Transdermal administration of Alzheimer’s drugs is an interesting and promising topic, which should be further elaborated on and studied [266,267].

## 6. Conclusions and Future Perspectives

AD is a progressive cortical neurodegenerative disease leading to the insidious onset of multiple cognitive disorders and gradual evolution over time. Existing therapeutic strategies are aimed towards slowing the progression of AD but are not curative approaches.

Strategies of prevention play an important role in the primary prevention of cognitive symptoms and are called for in this disease, where initial lesions form at an early preclinical stage and progress insidiously for years. Drug delivery to the CNS is a very complex process and challenging to ensure that the brain absorbs the full benefit of the drugs. In this review, new strategies to improve access to the therapeutic drug to the CNS have been analyzed and discussed, including liposomes and exosomes, shown to be effective drug delivery systems for the brain. Choosing the best pathway of medications and liposomes and exosomes to the brain is a very important topic in order to have the best efficiency as well as the better targeting zone in the brain. Among the different methods used in the literature, the IN method shows a better absorption efficiency because of the speed of the flow of drugs or NP to the brain. IN administration of liposomes is promising for the treatment of AD by virtue of its potential to facilitate molecule penetration across the BBB, better bioavailability, and efficacy by protecting the drug from peripheral degradation.

Other applications that also need to be considered include the use of gene editing for AD treatment [267,268], as well as antibody therapy targeting Aβ and tau proteins [269]. These topics have been discussed in other reviews as cited. Amyloid and aggregate clearance systems in non-human organisms such as yeast may also bring new insight into potential anti-Alzheimer therapeutics [270,271,272]. The combined use of strategies to reduce the amyloid burden and tau-protein aggregation with efficient delivery to brain tissues could be of potential benefit to AD patients.

## Figures and Tables

**Figure 1 ijms-23-13954-f001:**
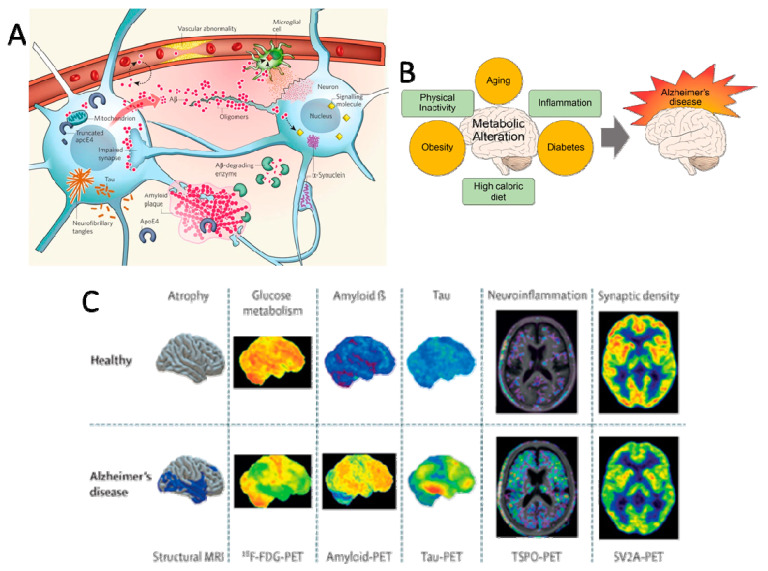
Alzheimer’s Disease spectrum. (**A**) Key players in the pathogenesis of Alzheimer’s disease (AD). Reprinted with permission from [12]. Copyright 2009, Nature Publishing Group. (**B**) Aging and metabolic diseases alteration of brain metabolism, which progressively causes AD. Reprinted from [65], Copyright 2017, Yonsei University College of Medicine. (**C**) Differential diagnosis of AD using neuroimaging biomarkers. Reprinted with permission [66]. Copyright 2009, Elsevier.

**Figure 2 ijms-23-13954-f002:**
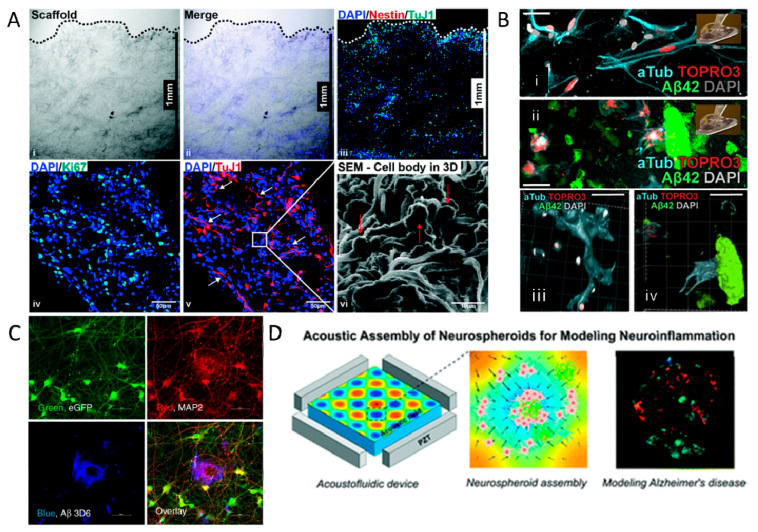
3D in vitro models of Alzheimer’s disease. (**A**) PLGA scaffold encapsulating iPSC-derived NPCs, shown by proliferation marker Ki67, undifferentiated cell marker Nestin, and neural differentiation marker TuJ1. (i–iii) Cross-section ofthe 3D microfiber scaffold showing cell infiltration, distribution and differentiation. (iv,v) cell proliferation and differentiation. (vi) SEM image of the scaffold cross-section. Reprinted with permission from [134]. Copyright 2020, The Royal Society of Chemistry. (**B**) Reduction of neural plasticity in neural stem cells as a result of Aβ plaques. Image (i,iii) indicate healthy neural stem cells, which can form new synaptic connections, shown in (iii). Image (ii,iv) have the Aβ plaques incorporated in the hydrogel and are unable to form new synaptic connections. Reprinted with permission from [135]. Copyright 2018, Elsevier. (**C**) 3D spheroids derived from human neural progenitor cells, displaying Aβ pathology (shown in blue). Green and Red channels depict the neuronal cells and differentiated neurons, respectively. Reprinted from [138]. Copyright 2020, Nature Portfolio. (**D**) Acoustofluidic platform to assemble cells and Aβ oligomers into neuro-spheroids representing AD. Reproduced with permission of The Royal Society of Chemistry [139]. (i) cross-section of the 3D microfiber scaffold after sectioning (dotted line indicates the top surface of the 3D scaffold); (ii,iii) cell infiltration, distribution and differentiation of iPSC-derived NPCs (8529 cell line) inside the 3D scaffold as assessed via staining for TuJ1 (green) and Nestin (red) markers on D13; (iv) cell proliferation assessed by the Ki67 (green) marker and (v) cell differentiation with neurite formation indicated by the TuJ1 (red) marker of hESC-derived NPCs on D13; nuclei were counterstained with DAPI (blue); (vi) SEM image of the scaffold cross-section showing cell morphology and attachment on microfibers at 2000× magnification.

**Figure 3 ijms-23-13954-f003:**
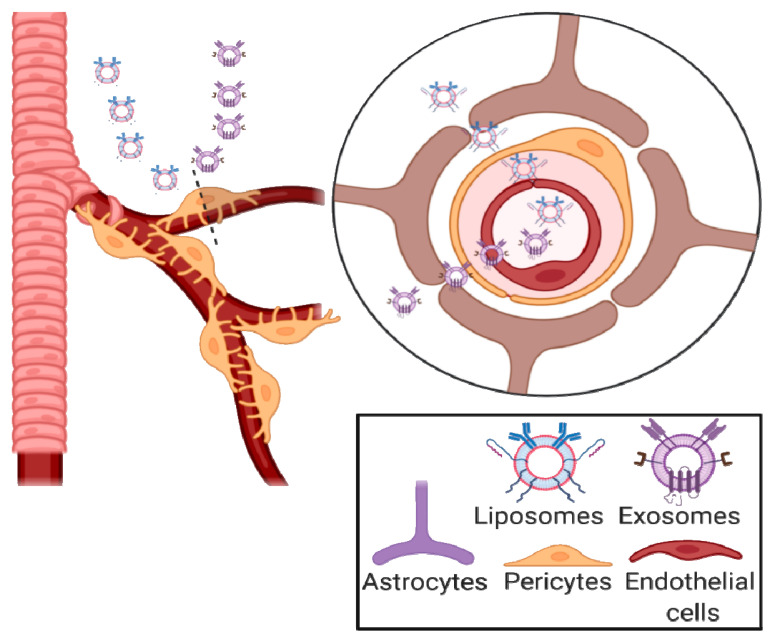
Schematic representation of liposomes, inorganic and polymeric nanoparticles crossing the BBB. Created with BioRender.com.

**Table 1 ijms-23-13954-t001:** Research models used for Alzheimer’s disease.

Type	Model	Key Findings	Ref.
**2D in vitro models**	hESC-derived neurons overexpressing *PSEN1*	Increased Aβ42/40 ratio due to depletion of Aβ40	[125]
	APP K724N mutated neurons from AD patients	Increased Aβ42/40 ratio due to depletion of Aβ40 and increased secretion of Aβ42	[126]
	hiPSC-derived astrocytes	Increased Aβ42/40 ratio in astrocytes is an important regulator of AD	[127]
	*PSEN1* ΔE9 mutated hiPSC-derived astrocytes	Increased Aβ42/40 ratio, ROS, increased cytokine release	[128]
	*PSEN1* M146L mutated hiPSC-derived astrocytes	Disturbed expression of astrocyte markers	[129]
	hiPSCs-derived neurons with *PSEN1* A246E mutation	Defective mitochondria have a key role in AD	[130]
	ReN immortalized stem cell line	Mutations in APP gene show accumulation of Aβ and phosphorylated tau	[131]
	PC12 immortalized cell line	GLP-1 neuroprotection and findings of Aβ toxicity	[132]
**3D in vitro models**	*PSEN1* A246E iPSC-derived neurons	Aβ aggregation without synthetic Aβ exposure or mutation induction	[133]
	iPSC-derived NPCs encapsulated in wet electrospun PLGA	Enhanced expression of Aβ42 and p-Tau	[134]
	NSCs encapsulated in starPEG-heparin-based hydrogels	Increased Aβ42 causes loss of neuroplasticity. System could allow for identification of therapeutic targets	[135]
	Induced NSCs in silk protein scaffold with HSV-1-induced AD	Aβ plaque formation, neuroinflammation, decreased functionality	[136]
	iPSCs-derived neuro-spheroids	Aβ aggregation; platform for testing of AD drugs	[137]
	3D human neural progenitor cells	Show the importance of reducing the Aβ42/40 ratio for amelioration of AD; accurate tau pathology	[138]
	Acoustofluidic platform for assembly of neurospheroids and Aβ plaques	High throughput screening platform to test drugs against Aβ plaques	[139]
	3D triculture of neurons, astrocytes, and microglial cells	Aβ aggregation, accumulation of p-tau, cytokine secretion	[140]
**In vivo models**	APP overexpressing mice	Aβ plaque formation, learning, and cognitive deficits after 6 months	[141]
	Aβ-GFP transgenic mice	Aβ is only able to form oligomers, thereby representing AD. Mice showed loss of memory, spine alterations, and increased p-tau levels.	[142]
	hTauP301L transgenic mice	Increased levels of phosphorylated tau, increased tau aggregation, neuronal loss	[143]
	T40PL-GFP transgenic mice, with the P301L 2N4R tau mutation	Increased levels of tau aggregation and tau pathology after 3 months	[144]
	ICV injection of Aβ oligomers	Memory loss in an ERK1/2-mediated fashion	[145]

## Data Availability

Not applicable.

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
