# Peer review of "Alzheimer’s Disease: Treatment Strategies and Their Limitations"

_ijms, 2022, doi:10.3390/ijms232213954_

Round 1

Reviewer 1 Report

The review is devoted to current therapeutical strategies of Alzheimer’s disease treatment and their limitations. The pathological basis of Alzheimer’s disease, models for study and treatment strategies including ways of drug delivery are discusses in detail.

However, there are certain issues which must be addressed.

The authors should at least mention in conclusions section the potential application of gene editing for Alzheimer’s treatment as well as application of antibody therapy with mentioning of most important targets for these therapies from the point of view of the authors.

Also the potential for the application of the knowledge related to amyloid- and aggregate clearance systems from yeast  (and other than human organisms) for the design of a new therapeutical approaches to target Aβ and Tau deposit should be at least mentioned in the section related to future prospectives of anti-Alzheimer’s therapy. Some insights for the discussion could be found for example here: DOI 10.1016/bs.adgen.2015.12.003; DOI 10.1021/acs.biochem.7b01285.  

Usage of amyloid- and protein aggregate-sequestering and destroying  proteins originating from other than human organisms in combination with approaches allowing to deliver these proteins to brain tissues could potentially benefit patients. Of course this requires further investigation.

Page 2. Line 52. Abbreviation ‘SAD’ should be explained.

The list of abbreviations used would benefit the paper.

Page 2. Line 76. What is ‘ND’? 

Page 2. Line 92, 98. What is ‘SNC’?

Page 2. Line 88. The sentence should be checked for a grammatical error.

Page 2. Line 94. What is ‘NL’? Later in the text similar abbreviation means nanoliposomes.

Page 23. Line 984, 985. There are spelling errors in names or articles cited.

Page 3. Lines 122-123. The sentence should be rewritten for clarity.

Page 3. Line 136. Abbreviation ‘NFT’ has been explained in previous part of the manuscript.

Page 4. Line 168. The sentence should be checked for a grammatical error.

Page 4. Line 195.  The sentence should be checked for a grammatical error.

Page 7. Line 319. The sentence should be checked for a grammatical error.

Page 10. Line 412. The title of the subsection misses number ‘2’ in front of letter ‘D’. The same comment is applied for the name of the next subsection (3.2.3).

The reference 116 is the same as the reference 117.

The paper can be accepted after minor revision.

Author Response

The review is devoted to current therapeutical strategies of Alzheimer’s disease treatment and their limitations. The pathological basis of Alzheimer’s disease, models for study and treatment strategies including ways of drug delivery are discusses in detail.

However, there are certain issues which must be addressed.

The authors should at least mention in conclusions section the potential application of gene editing for Alzheimer’s treatment as well as application of antibody therapy with mentioning of most important targets for these therapies from the point of view of the authors.

Also the potential for the application of the knowledge related to amyloid- and aggregate clearance systems from yeast  (and other than human organisms) for the design of a new therapeutical approaches to target Aβ and Tau deposit should be at least mentioned in the section related to future prospectives of anti-Alzheimer’s therapy. Some insights for the discussion could be found for example here: DOI 10.1016/bs.adgen.2015.12.003; DOI 10.1021/acs.biochem.7b01285.  

Usage of amyloid- and protein aggregate-sequestering and destroying  proteins originating from other than human organisms in combination with approaches allowing to deliver these proteins to brain tissues could potentially benefit patients. Of course this requires further investigation.

We have included this information in the conclusions and perspectives, along with citations of reviews that cover these topics in detail. 

Page 2. Line 52. Abbreviation ‘SAD’ should be explained.

The changes has been incorporated into the text.

The list of abbreviations used would benefit the paper.

We have added a list of abbreviations.

Page 2. Line 76. What is ‘ND’? 

ND is abbreviation of Neurodegenerative diseases. We have added (ND) in line 40.

Page 2. Line 92, 98. What is ‘SNC’?

It’s CNS, we have modified into the text.

Page 2. Line 88. The sentence should be checked for a grammatical error.

The changes has been incorporated into the text.

Page 2. Line 94. What is ‘NL’? Later in the text similar abbreviation means nanoliposomes.

It’s nanoliposomes, we have added into the text.

Page 23. Line 984, 985. There are spelling errors in names or articles cited.

The changes has been incorporated into the references.

Page 3. Lines 122-123. The sentence should be rewritten for clarity.

The changes has been incorporated into the text.

Page 3. Line 136. Abbreviation ‘NFT’ has been explained in previous part of the manuscript.

The changes has been incorporated into the text.

Page 4. Line 168. The sentence should be checked for a grammatical error.

The changes has been incorporated into the text.

Page 4. Line 195.  The sentence should be checked for a grammatical error.

The changes has been incorporated into the text.

Page 7. Line 319. The sentence should be checked for a grammatical error.

The changes has been incorporated into the text.

Page 10. Line 412. The title of the subsection misses number ‘2’ in front of letter ‘D’. The same comment is applied for the name of the next subsection (3.2.3).

The changes has been incorporated into the text.

The reference 116 is the same as the reference 117.

 The changes has been done.

The paper can be accepted after minor revision.

We thank the reviewer for the feedback. We have made the corrections requested. The changes have been incorporated into the text and the references and we hope it now makes everything clear.

Reviewer 2 Report

“Alzheimer's Disease: Treatment Strategies and Their Limitations” by Elodie Passeri, Kamil Elkhoury, Margaretha Morsink, Kerensa Broersen, Michel Linder, Ali Tamayol, Catherine Malaplate, Frances T. Yen  and Elmira Arab-Tehrany

This review article by Passeri et al is an extensive documentation of the various treatment strategies that are in use for Alzheimer’s disease along with the limitations of each.  It starts with a nice introduction to the disease and its pathophysiology, followed by causative and risk factors, research models of the disease, delivery methods, etc. They also explore the various drug delivery systems as well as drug administration systems and provide their opinion on each.

Overall, the manuscript is well written and very extensive.  References have been cited for most of the work that they have described.  The language is good and there is a good flow in the manuscript.  I have just seen a few minor things that need to be corrected before accepting the paper for publication.  These places are highlighted in the attached file and the authors need to make these small corrections.  When full sentences or a few sentences are highlighted, it indicates that a reference on that would be useful.  Rest are for typographical errors or small grammatical errors that can be corrected.

Author Response

“Alzheimer's Disease: Treatment Strategies and Their Limitations” by Elodie Passeri, Kamil Elkhoury, Margaretha Morsink, Kerensa Broersen, Michel Linder, Ali Tamayol, Catherine Malaplate, Frances T. Yen  and Elmira Arab-Tehrany

This review article by Passeri et al is an extensive documentation of the various treatment strategies that are in use for Alzheimer’s disease along with the limitations of each.  It starts with a nice introduction to the disease and its pathophysiology, followed by causative and risk factors, research models of the disease, delivery methods, etc. They also explore the various drug delivery systems as well as drug administration systems and provide their opinion on each.

Overall, the manuscript is well written and very extensive.  References have been cited for most of the work that they have described.  The language is good and there is a good flow in the manuscript.  I have just seen a few minor things that need to be corrected before accepting the paper for publication.  These places are highlighted in the attached file and the authors need to make these small corrections.  When full sentences or a few sentences are highlighted, it indicates that a reference on that would be useful.  Rest are for typographical errors or small grammatical errors that can be corrected.

We thank the reviewer for your attention. We have made all the corrections requested into the text.
